# Construction of a Chikungunya Virus, Replicon, and Helper Plasmids for Transfection of Mammalian Cells

**DOI:** 10.3390/v15010132

**Published:** 2022-12-31

**Authors:** Mayra Colunga-Saucedo, Edson I. Rubio-Hernandez, Miguel A. Coronado-Ipiña, Sergio Rosales-Mendoza, Claudia G. Castillo, Mauricio Comas-Garcia

**Affiliations:** 1Sección de Genómica Médica, Centro de Investigación en Ciencias de la Salud y Biomedicina, Universidad Autónoma de San Luis Potosí, San Luis Potosí 78210, Mexico; 2Laboratorio de Células Troncales Humanas, Coordinación para la Innovación y Aplicación de la Ciencia y la Tecnología, Facultad de Medicina, Universidad Autónoma de San Luis Potosí, San Luis Potosí 78210, Mexico; 3Sección de Microscopía de Alta Resolución, Centro de Investigación en Ciencias de la Salud y Biomedicina, Universidad Autónoma de San Luis Potosí, San Luis Potosí 78210, Mexico; 4Sección de Biotecnología, Centro de Investigación en Ciencias de la Salud y Biomedicina, Universidad Autónoma de San Luis Potosí, San Luis Potosí 78210, Mexico; 5Facultad de Ciencias Químicas, Universidad Autónoma de San Luis Potosí, San Luis Potosí 78210, Mexico; 6Facultad de Ciencias, Universidad Autónoma de San Luis Potosí, San Luis Potosí 78295, Mexico

**Keywords:** Chikungunya virus, alphavirus DNA vectors, replicon plasmid, helper plasmid

## Abstract

The genome of Alphaviruses can be modified to produce self-replicating RNAs and virus-like particles, which are useful virological tools. In this work, we generated three plasmids for the transfection of mammalian cells: an infectious clone of Chikungunya virus (CHIKV), one that codes for the structural proteins (helper plasmid), and another one that codes nonstructural proteins (replicon plasmid). All of these plasmids contain a reporter gene (mKate2). The reporter gene in the replicon RNA and the infectious clone are synthesized from subgenomic RNA. Co-transfection with the helper and replicon plasmids has biotechnological/biomedical applications because they allow for the delivery of self-replicating RNA for the transient expression of one or more genes to the target cells.

## 1. Introduction

Chikungunya virus (CHIKV) belongs to the *Togaviridae* family and genus *Alphavirus,* and it is also known as an Arbovirus (Arthropod-borne virus) because it is transmitted by mosquitos from the genus *Aedes*. According to the European Centre for Disease Prevention and Control, in 2022, there were 338,592 confirmed cases and 70 deaths associated with CHIKV. The majority of cases were reported in Brazil (240,344), India (93,113), Guatemala (1435), Thailand (775), and Malaysia (662). A meta-analysis of 44 studies worldwide that included 51,599 people from 29 countries revealed that up to 25% of the population could be seropositive to CHIKV. The region with the highest prevalence was Southeast Asia, with 42%, and the lowest was the Eastern Mediterranean, with 2% [1]. CHIKV has a positive-sense, single-stranded RNA genome (gRNA) that is encapsulated in an T = 4 icosahedral virion. The virion is composed of a nucleocapsid containing 240 monomers of capsid protein, one copy of the gRNA, a lipid bilayer, and eighty trimers of a heterodimer of the glycoproteins E2/E1. The capsid proteins and glycoproteins interact with each other and assemble into a virion with icosahedral symmetry (both at the core and virion level) with a diameter between 50 and 70 nm [2]. The gRNA (11.5 kb) is divided into the 5′ and 3′UTR, an intergenic region, and two open reading frames (ORF1 and 2). The ORF1 encodes four nonstructural proteins (nsP1234), which are part of the RNA-dependent RNA polymerase complex (RdRp) and contains the sequence for a putative packaging signal. The ORF2 encodes the structural proteins C, E3, E2, E1, TF, and 6K [2,3]. An advantage of the gene organization of Alphaviruses is that it allows for the replacement of the ORF2 by one or more non-viral sequences, giving rise to a self-replicating RNA called replicon that amplifies the gene of interest but does not produce viral particles. In addition, deletion of the ORF1 results in a defective helper RNA that, under certain circumstances, can generate structural proteins, but it is unable to self-replicate [4,5,6]. When the helper RNA is in a mammalian expression plasmid and depending on how the ORF1 was deleted, the amplification of the helper RNA can depend either on the RdRp (Replicon-dependent helper) or on the cellular polymerases (Replicon-independent helper) [7,8,9,10,11]. In other words, deletion in the ORF2 results in a self-replicating RNA that does not give rise to virions (i.e., replicon RNA), while deletion of the ORF1 results in an RNA that will produce virus-like-particles, but it cannot be self-amplified (i.e., helper RNA).

The need to generate new therapies against autoimmune, infectious, and non-communicable diseases has resulted in the development of new strategies for gene expression. Most systems that allow for the delivery and expression of heterologous genes in cell cultures, animal models, and humans are based on viral vectors derived from DNA viruses and retroviruses. The most common are Adenovirus, Lentivirus, Herpesvirus, and Adeno-associated viruses [12,13,14,15,16,17]. However, while these vectors are extremely promising, they have some disadvantages caused by their limited tropism and the high prevalence of similar viruses, resulting in pre-existing immunity that decreases the efficacy of the viral vector. In addition, in some cases, there may be a risk (although of low probability) of integration of the viral genome into the host genome [18]. These are some reasons that have motivated the development of alternative gene delivery systems; the best examples are micelles and lipid vesicles containing mRNAs [19]. A disadvantage of these lipid/mRNA-based systems is the infrastructure required for transport and long-term storage before reconstitution (e.g., ultra-low temperature freezers at −80 °C) [20].

It is in this context that the generation of a replicon/helper system derived from an Alphavirus is an appealing approach. For example, the genome can be easily modified to generate single-round viral vectors [21,22,23,24]. This system can be used to produce viral particles that are structurally identical to the parental virus that contain a gene of interest instead of structural proteins [11,25]. The structural proteins account for about 4.6 kilobases; hence, they could be replaced by several genes, which can be separated into several individual proteins by using several subgenomic promoters and/or by fusing them to peptides, such as P2A [5,26,27]. Further, CHIKV has a very wide tropism because it can use multiple receptors and cellular factors, such as Actin gamma 1, collagen type I-alpha-2, and PTPN2, PHB1, Mxra8, DC-SIGN, TIM-1, and glycosaminoglycans [28,29,30,31,32,33]. Furthermore, given the fact that the Old and New-World alphaviruses have different tropisms, it is possible to exchange the glycoproteins of an alphavirus that produced dengue fever, such as disease (e.g., CHIKV) for one that results in encephalitis (e.g., Venezuelan equine encephalitis virus) and, thus, tropism of the viral vector [34]. It is also important to point out that the seroprevalence in the human population against this virus in places within non-tropical climates is low compared to other viruses [35]. In cases where the seroprevalence against a particular Alphavirus was to be high, the glycoproteins of this viral vector could be exchanged for ones of another alphavirus with the same tropism but with low seroprevalence in such a region. Nonetheless, the seroprevalence of CHIKV is mostly limited to the tropical areas of the Americas, the Indian subcontinent, and southeast Asia; the rest of the world has a seroprevalence between 0 and 2% [1].

One disadvantage of most Alphavirus helper/replicon systems is that, in most cases, single-round infectious particles are generated by the co-transfection or co-electroporation of in vitro transcribed RNAs, which is not optimal for the large-scale production of viral vectors. Therefore, there is a need to generate CMV-driven single-round infectious CHIKV particles that could be affordable compared to those that are based on transfection with RNA.

Over the course of more than two decades, alphaviruses have been studied to understand how their genome is organized, how they replicate and assemble, and their infection mechanism; however, there is still no approved vaccine or specific treatment against this pathogen [36,37,38,39,40,41]. There are several limitations that have lagged behind the development of alphavirus-based therapies or vaccines. For example, using electroporation to introduce in vitro transcribed viral RNAs into insect or mammalian cells is expensive and impractical for massive administration. Further, the RNA can be easily degraded and scaling up its production can be challenging. Nonetheless, Alphavirus replicon/helper systems represent a promising approach for immunization and protection against pathogenic viruses and gene therapy [17,42,43,44,45]. However, in order for this technology to become a reality, we need to move the helper/replicon systems from using in vitro transcribed RNAs to plasmids for mammalian cell expression (e.g., pVax1 or pcDNA3.1, which in both cases generate mRNAs using the cellular nuclear machinery thanks to the presence of a cytomegalovirus (CMV) RNA polymerase II promoter and simian vacuolating virus 40 (SV40) and β-globin polyA signals). In fact, it has been shown that plasmid-based replicon vectors produce higher levels of encoded heterologous proteins compared to conventional DNA vaccines and provide an affordable platform [46,47,48,49]. It is important to point out that there are several plasmids that contain the full-length CHIKV genome either under an SP6 or T7 in vitro transcription promoter [50,51,52,53,54] or under a CMV promoter [54,55,56,57]. Moreover, most replicon/helper systems require transfection or electroporation of in vitro transcribed RNA rather than transfection with a CMV-driven plasmid [7].

Here, we report the generation of a plasmid for expression in mammalian cells that contains the full-length genome of the attenuated strain 181/25 of the Chikungunya virus under the CMV promoter and two transcription terminators and polyA signals. In this plasmid, the viral 3′ UTR and polyA were maintained intact by inserting the hepatitis delta virus ribozyme. Further, this clone expresses the mKate2 protein as a reporter gene. Transfection of HEK-293T cells with this plasmid results in particles able to infect Vero E6 and HEK-293T cells. Ultrastructural analysis of the infected cells reveals virions and type-I and -II cytopathic vesicles identical to those observed with the wild-type virus. Furthermore, we generated two more expression plasmids for mammalian cells that encode for either the ORF1 (replicon) or the ORF2 (Replicon-independent helper); thus, we were able to generate a single-round infectious virus. This study demonstrates that Chikungunya DNA vectors can be easily modified to have the potential to study assembly (i.e., by using the replicon-independent helper plasmid), how this virus replicates its genome (i.e., by using the replicon plasmid), to generate single-round infectious particles to understand the infectious cycle, to determine the neutralizing activity of antibodies, or for gene delivery.

## 2. Materials and Methods

### 2.1. Plasmid Constructions

The original plasmid used to generate helper and replicon plasmids contained the full-length genome of the attenuated strain 181/25 of the Chikungunya virus in a vector for in vitro transcription with the SP6 RNA polymerase (SP6-CHIKV). First, the original plasmid was modified to change the SP6 transcription promoter to a BamHI restriction site by site-directed mutagenesis using primers SP6toBamHI-F and SP6toBamHI-R (see Appendix A) and a polymerase Phusion HS II (ThermoFisher Scientific, Waltham, MA, USA). The PCR product was digested with DpnI (ThermoFisher Scientific) and directedly used to transform NEB 5-α-competent cells (New England Biolabs, Ipswich, MA, USA). The resulting colonies were grown in LB media with ampicillin (Caisson Laboratories, North Logan, UT, USA), and the plasmids were purified using the alkaline lysis method [58]. The modified plasmid, as well as the pVax1 plasmid (ThermoFisher Scientific, Waltham, MA, USA), were digested with NotI and BamHI (New England Biolabs, Ipswich, MA, USA) enzymes. The digested pVax1 was treated with alkaline phosphatase (New England Biolabs, Ipswich, MA, USA), then both plasmids were ligated with a T4 DNA ligase (New England Biolabs, Ipswich, MA, USA) and were transformed into NEB 5α cells. The resulting colonies were grown in LB media with kanamycin (Caisson Laboratories, North Logan, UT, USA).

The replicon-independent helper plasmid (pVax-Help) was generated by deleting the non-structural proteins 1–4 (161–7510 pb) from pVax-CHIKV. First, the plasmid was linearized by inverse PCR using the primers DelNsP4-F and DelNsP1-R (Appendix A). Then, the plasmid was re-circularized using the KLD kit (New England Biolabs, Ipswich, MA, USA) and directly transformed into NEB 5α cells. The resulting colonies were grown in LB media with kanamycin.

The replicon plasmid pVax-Rep was generated by two inverse PCR/re-circularization consecutive protocols as described above using the pVax-CHIKV plasmid as a template. First, the capsid gene was deleted (7571–8355 pb) by inverse PCR using the primers DelCPE3-F and DelCPE3-R (Appendix A), re-circularized with the KLD kit, and transformed into NEB 5α cells. The resulting plasmid was used as a template to delete the E3 to E1 genes (9058–12,072 pb) using the primers DelFMtoE1-F and DelFMtoE1-R (Appendix A), ligated with the KLD kit, and transformed into NEB 5α cells.

The full-length CHIKV genome and the replicon sequences were moved from the pVax vector to a pACNR1811 plasmid [59]. The viral sequence was inserted between the CMV promoter and the hepatitis delta virus (HDV) sequence. The original viral polyA was conserved during this step. This mutagenesis was performed using the In-Fusion Cloning Kit method (Clontech, Mountain View, CA, USA). The pACNR1811 lineal vector was inverse-PCR amplified using the primers ICD-F and ICD-R. Then, the viral sequences were amplified by PCR with the primers Chik-F and Chik-R (Appendix A). All PCR products were DpnI-digested, column-purified, UV-Vis-quantified, and ligated following the vendor’s protocol. The resulting plasmids were called pACNR-CHIKV and pACNR-Rep.

All plasmids were SANGER-sequenced (LANBAMA-IPICyT, Mexico). The annotated sequences are in the Appendix A and the electronic files are freely available upon request.

### 2.2. Cell Culture Conditions

HEK-293T (CRL-3216) cells were obtained from American Type Culture Collection (ATCC, Manassas, VA, USA). HEK-293T cells were grown and maintained in Dulbecco’s modified Eagle medium (DMEM; Corning, Manassas VA, USA). The culture medium was supplemented with 10% fetal bovine serum (FBS; Gibco, Thermo Fisher Scientific, Waltham, MA, USA). Vero E6 cells (CRL-1586) (ATCC, Manassas, VA, USA) were grown in DMEM (Corning, Manassas VA, USA) and supplemented with 2% FBS (Gibco, Thermo Fisher Scientific, Waltham, MA, USA). All cell lines were grown in a 5% CO_2_ atmosphere at 37 °C using culture tissue-treated plates and flasks (Corning, Manassas VA, USA).

### 2.3. Plasmid Transfection

Infectious CHIKV particles were produced by transfecting HEK-293T cells with the pACNR-CHIKV plasmid. The single-round infectious particles and virus-like-particles (VLPs) were produced in HEK-293T cells (producer cells) by either co-transfecting them with the pACNR-Rep and pVax-Help plasmids or just pVax-Help, respectively. The cells were seeded in a 6-well dish at a cell density of 250,000 cells/well. Twenty-four hours later, the cells were transfected with lipofectamine 3000 (Invitrogen, Carlsbad, CA, USA) reagent. The viral particles and VLPs were collected at 24 and 48 h post-transfection (h.p.t.), centrifuged at 1000 RFC for 10 min, and frozen at −80 °C. The expression of mKate2 and cell morphology were analyzed by fluorescence microscopy Lionheart FX Automated Microscope (BioTek Instruments, Winooski, VT, USA) using a Texas RED and DAPI filters, as well as a high-contrast phase mode. All the DNAs used for transfection were purified with the kit MaxiPrep (Promega Corporation, Fitchburg, WI, USA).

#### Expression Kinetics of mKate2 Protein Reporter with Chikungunya Plasmid

HEK-293T cells were transfected with 0.5 µg and 1 µg of pACNR-CHIKV following the above-described protocol. The fluorescence signal was measured in a fluorescence microscopy Lionheart FX Automated Microscope using a Texas RED filter (termed as mKate2 in the figures) for 96 h every 12 h.

### 2.4. Entry Assay

The entry of the viral particles and VLPs was performed by overlaying 165 μL of the supernatant of the producer cells on HEK-293T cells seeded in a 12-well dish at a cell density of 100,000 cells/well, which were previously washed to remove the FBS. Two hours post-infection (h.p.i.), 1 mL of fresh media was added to the cells, and 24 h.p.i., the expression of mKate2 and Hoechst stain was analyzed in a fluorescence microscopy Lionheart FX Automated Microscope.

### 2.5. CHIKV Infection and Viral Quantification by Plaque Assay

Vero E6 cells were infected with the supernatants obtained from cultures subjected to the transfection previously described with pACNR-CHIKV. The viral progeny virus was collected at 24 and 48 h post-infection and filtered through a 0.22 μM syringe filter and frozen at −80 °C. The quantification of the virus was performed by overlaying 450 μL of the supernatant of the infected cells on Vero E6 cells seeded in a 6-well dish at a cell density of 400,000 cells/well, which were previously washed to remove the serum. Two hours h.p.i., 3 mL of 1% Carboxymethylcellulose (Sigma-Aldrich, Saint Louis, MO, USA) was added to the cells and incubated. After 4 days, the cells were stained with crystal violet solution for visualization and counting of plaques [60]. The supernatant of mock-infected cells was used as a negative control.

### 2.6. Ultra-Structural Analysis by Thin-Section Transmission Electron Microscopy

Monolayers of cells were either infected with CHIKV at a MOI of 1 or co-transfected with pACNR-Rep and pVax-Help. Then, 24 h-post-infection and 48 h.p.t., the cells were fixed with a solution of 4% formaldehyde, 0.05% picric acid, and 0.1 M sodium cacodylate, pH 7.4. The cells were processed following the protocol from Tobin et al. 1996 [61]. The samples were washed with EMBed 812 resin and polymerized for 48 h at 55 °C. The resin was cut with an EMUC7 ultramicrotome into 70 nm-thick slices, which were mounted on copper TEM grids and stained with uranyl acetate and lead citrate. Samples were analyzed on a JEM-2100 transmission electron microscope (JEOL Ltd., Akishima, Tokyo, Japan) at 200 kV using a Gatan OneView 4K camera.

## 3. Results

### 3.1. Generation of Viral, Replicon, and Helper CHIKV Plasmids for Expression in Mammalian Cells

The plasmid SP6-CHIKV is used to produce CHIKV via electroporation of in vitro transcribed RNA with SP6 polymerase; hence, we decided to transfer the CHIKV genome from an SP6-based vector to a CMV-driven plasmid for mammalian expression (i.e., pVax1). Since the pVax1 plasmid has its own polyadenylation signal, the viral polyA tail was removed. Then, we mutated the pVax-CHIKV plasmid to produce two more plasmids: the helper plasmid was generated by deleting non-structural proteins (nsP1234) and the replicon plasmid was obtained by deleting capsid protein and E1-E2-6K-E1 proteins (Figure 1A).

Transfection of HEK-293T cells with pVax-CHIK (data not shown) and pVax-Rep (Appendix A) did not result in the detection of the reporter gene mKate2. However, transfection with pVax-Help results in the expression of mKate2, as seen by the red signal in the mKate2 channel (Appendix A). The transcription of pVax-Help mRNA depends on cellular polymerase, while mRNA transcription of pVax-CHIKV and pVax-Rep depends on cellular polymerase and viral RNA-dependent RNA polymerase (RdRp). In order to test if the polyadenylation signal from the pVax affected the functionality of the viral sequences, the CHIKV genome and the replicon RNA were moved from pVax to the plasmid pACNR, which contains an HDV ribozyme after the viral polyA tail (Figure 1A). Figure 1B shows that the transfection of HEK-293T cells with pACNR-CHIKV resulted in the expression of the reporter gene mKate2 (red cells) and that most cells are mKate2-positive. This signal indirectly shows expression of the viral proteins from the subgenomic RNA (i.e., SGP in Figure 1A). The kinetics of the transfection was monitored by measuring the number of mKate2-positive cells using an inverted fluorescence microscope. Figure 1C shows that the transfection with 1 μg resulted in a higher expression of mKate2 at early times (before 48 h.p.t.) compared to the cells transfected with 0.5 μg. However, at around 48 h.p.t., the amount of mKate2-positive cells was higher in the cultures transfected with 0.5 μg of DNA than 1.0 μg. After 72 h.p.t., the total number of cells and mKate2-positive cells decreases and the integrity of the cell monolayer disappears.

### 3.2. Viral Production and Analysis of Infection by Microscopy

To corroborate that the particles obtained by transfecting cells with pACNR-CHIKV are infectious, the supernatant of the transfected cells was used to infect HEK-293T and Vero E6 cells. Figure 2A shows the fluorescence micrographs of HEK-293T infected to MOI of 1; the red signal indicates the cells that are transfected and the blue signal indicates the cell nuclei. This data demonstrate the expression of the reporter gene located in the subgenomic RNA. Figure 2A also shows the same for Vero E6 cells; however, by comparing the number of mKate2-postive cells compared to the number of nuclei, it seems that the infection efficiency in Vero E6 cells is lower than in HEK-293T (Figure 2A). Nonetheless, Figure 2B demonstrates that the infected Vero E6 cells exhibited the cytopathic effects characteristic of CHIKV infection; the cells change from barely visible by light microscopy to round cells that refract light. Figure 2C shows the typical plaques from this infection in Vero E6 cells. From the plaque assays in Vero E6, we determined that the cumulative production of CHIKV was 1.77 × 10^7^ PFU/mL and 2.21 × 10^7^ PFU/mL at 24 h.p.i. and 48 h.p.i., respectively (Figure 2D). The data suggest that transfection of mammalian cells, such as the HEK-293T with the pACNR-CHIKV plasmid, results in infectious virions that can infect HEK-293T and Vero E6 cells.

Figure 3A shows viral particles, released from the cell as well as budding particles, with their characteristic double membrane and protrusion from the E2/E1 proteins (see triangle). This figure also shows budder cores at the plasma membrane where they interact with the glycoproteins so that the virion can be fully assembled and can then bud out. Figure 3B shows an accumulation of cores in the cytoplasm near the endoplasmic reticulum. Figure 3C shows the core particles before interacting with the glycoproteins that are not budding yet. Finally, Figure 3D shows the classic type-II cytopathic vacuoles. In this vacuole, the viral particles surround the central vacuole, while traveling to the plasma membrane. All the virions present the characteristic morphology of CHIKV with an approximate diameter of 50 nm (Appendix A).

### 3.3. The Replicon and Helper Vector System Generates Single-Round Infectious Particles with Gene Delivery Capability

The next step was to test whether the plasmid pACNR-Rep results in the expression of the fluorescent gene mKate2 (Figure 4A). This plasmid cannot give rise to infectious particles because it does not contain structural genes. Figure 4B shows fluorescence micrographs (mKate2), phase contrast, and a merge of them of HEK-293T cells transfected with pACNR-Rep, pVax-Help, and pACNR-Rep/pVax-Help. The expression of the reporter gene, generated from the subgenomic RNA, shows that both plasmids are functional. Subsequently, HEK-293T cells were co-transfected with both plasmids; this resulted in the fluorescence intensity and the number of co-transfected fluorescent cells being higher than when the cells were transfected with only one of the two plasmids.

To demonstrate the functionality of the replicon and helper vector system, the supernatants of the co-transfected producer cells were removed within 24 h, the cells were washed, and a fresh culture medium was added. Figure 5A shows that the supernatant of cells co-transfected with pACNR-Rep and pVax-Help results in the expression of the reporter gene mKate2 after 48 h. However, the yield of the infection is extremely low compared to the number of infected cells when using the full-infection clone. Finally, electron transmission microscopy studies were performed on ultrathin sections of cells co-transfected with pACNR-Rep and pVax-Help. Figure 5B shows the presence of type-I cytopathic vacuoles that contain small vesicles in their interior. Figure 5C shows a representative micrograph showing virions budding from the producing cell, while Figure 5D shows virions released from the cell with an approximate diameter of 66 nm.

## 4. Discussion

The generation of infectious viral particles belonging to the *Togaviridae* family is usually achieved by electroporation of cultured cells with in vitro transcribed RNA [7,50,52,53,54]. Although this methodology is useful for the study of the biology of these viruses, it is not suitable for their use as viral vectors and/or for large-scale production of viral antigens (e.g., replication-deficient or inactivated viruses). The first aim of this project was to create an infectious clone of CHIKV in a CMV-driven plasmid for mammalian expression to eliminate the need for using in vitro transcripts. In our study, neither the CHIKV genome nor the replicon genome in the pVax plasmid results in the generation of the subgenomic RNA. This is consistent with the low levels of expression of another CMV-driven replicon where the viral polyA is not maintained intact [62]. The plasmid pVax-Helper resulted in the expression of the reported gene, which is consistent with the fact that the transcription of this mRNA does not depend on the CHIKV RdRp. The lack of any signal from the reporter gene in the cells transfected with pVax-CHIKV and pVax-Rep suggests that the β-globin polyadenylation termination signal introduces extra nucleotides that alter the structure of the 3′UTR region. This region is known to be essential for the RNA-dependent RNA polymerase (RdRp) to recognize the viral RNA [63]; mutations within this region could prevent the RdRp from recognizing the viral sequence and, thus, inhibit viral replication. To test this hypothesis, the 3′UTR was corrected by the addition of HDV ribozyme site after the viral polyA sequence and before the polyadenylation and termination signals. This approach has been used in previous studies with CHIKV, Kunjin, Venezuelan equine encephalitis, and Sindbis virus because it excises the last nucleotide of the viral sequence from the first nucleotide of the ribozyme [46,54,55,56,64,65]. As expected and in line with the findings reported by Suzuki [56], the plasmids pACNR-CHIKV and pACNR-Rep expressed the mKate2 gene in transfected HEK-293T cells. Our results, along with those for CHIKV and other Alphaviruses, indicate that only the intact viral 3′UTR allows the RdRp to synthesize the subgenomic RNA from which the reporter gene is translated. Furthermore, these results along with those from Suzuki demonstrate that HEK-293T and Vero E6 cells are susceptible and permissible to CHIKV infection. The susceptibility of this cell line is most likely due to the presence of the cellular PHB1 in HEK-293T cells and TIM-1 in Vero E6 cells [30,66].

The presence of the reporter protein that translates in *cis* with the viral proteins allows us to directly evaluate, in real time, the replication of RNA and indirectly the translation of structural proteins. After 72 h.p.t., the integrity of the cell monolayer was lost, suggesting that the cells can no longer support viral replication, most likely due to cell death. Furthermore, at this time point, the media need to be replaced otherwise they becomes acidic, which, in turn, results in particles that are not infectious. This is consistent with the fact that CHIKV enters cells by endocytosis and, thus, it is possible that an acid medium changes the conformation of the E1 and E2 proteins of the extracellular particles to the one that is required for the virion to escape the endosome [67]. Further, we observed that a lower amount of DNA gives a higher yield of transfection. In our hands, this phenomenon occurs for almost any transfection with Lipofectamine 3000^®^; it is possible that high amounts of this reagent could be cytotoxic.

The infection in HEK-293T results in greater cell death than in Vero E6 cells. One possibility for this is that the latter does not have an interferon response. In fact, we observed the same effect with the Zika virus [60]. In addition, these particles generate, in Vero E6 cells, a cytopathic effect and plaques identical to those reported for this virus [68,69]. The particles observed by thin-section TEM near the plasma membrane have an approximate diameter of 50 to 60 nm, consistent with previous reports [70]. Although this variation has been previously observed [70], it is important to point out that we measured the diameters of the particles by thin-section TEM and not by negative-staining TEM of purified particles. Therefore, the virions that were not sectioned at the center of the particle will have a diameter smaller than expected. Recently, it was demonstrated that the mechanism of entry is clathrin-mediated in Vero cells (6 h.p.i.) [71]. Nonetheless, we were not able to observe the endocytosis of viral particles via thin-section TEM. This might be because the cells were fixed at 24 h.p.i., rather than 6 h.p.i. Furthermore, at 24 h.p.i. CHIKV, we were able to observe CPV-II and viral factory, supporting the idea that at this time, almost all the entry processes have already occurred and genome replication is taking place.

The generation of replicon and replicon/helper systems derived from Alphaviruses is not novel. However, for CHIKV, this system was generated by using plasmids for in vitro transcription rather than for mammalian expression [7,62,72,73]. Gläsker and co-workers generated a CHIKV replicon/helper system for transfection or electroporation with in vitro transcribed RNA rather than a CMV-driven plasmid [7]. However, this is not the only difference between the system presented here and the one from Gläsker. They generated the structural proteins by using two helper RNAs, while we only needed one plasmid. Further, the reporter gene they used was Gaussia luciferase, which is secreted into the supernatant instead of remaining intracellularly, as in our case.

On the one hand, the use of Gaussia luciferase has the advantage that this reporter system allows for the detection of lower amounts of the reporter gene than with most intracellular fluorescent protein. On the other hand, because Gaussia luciferase is secreted, it does not allow one to determine either the number of infected cells or to identify which cells are infected. In other words, by using mKate2 instead of Gaussia luciferase, we can determine the percentage and identity of infected cells by fluorescence microscopy. Utt et al. further reported a replicon system [62]; however, to detect the activity of the RdRp, they have to transfect cells with either two RNAs or CMV-driven plasmids: one molecule code for the RdRp, while the other contains a reporter gene flanked by the regulatory sequences of the CHIKV RdRp. Unlike the system of Utt, our CMV-driven replicon plasmid generated here has a *cis* reporter gene and, hence, we only need to use one plasmid rather than two. Furthermore, the CMV-driven plasmid from Utt is barely functional, most likely because they did not fix the 3′UTR.

The fact that the infection of HEK-293T cells with the supernatant of the co-transfected cells at 48 h.p.t. results in the expression of the reporter gene confirms that the single-round infection particles packaged the replicon RNA. The presence of type-I cytopathic vacuoles indicates that the cellular localization of the non-structural proteins is the same as when an infectious clone is used [71]. Altogether, these data show that cells doubly transfected with pACNR-Rep and pVax-Helper result in the production of particles that are identical to CHIKV and that contain (at least some of them) the replicon RNA. However, the efficiency of infection with the single-round infectious particles produced in the doubly transfected cells was low compared to the infectious clone. This could be explained by the following scenario: in the producer cells, all the single-round infectious particles that are released prior to the collection time can enter non-transfected cells (as these cells are susceptible to CHIKV). Nonetheless, unlike with the infectious particles, the entry of single-round infectious particles does not result in the production of viral particles. Gläsker evaluated an alternative to resolve this problem and designed a replicon and helper system vectors, in which the synthesis of the helper RNA depends on the CHIKV RdRp [7]. It is important to point out that the single-round infectious particles reported by Gläsker use Gaussia luciferase as a reporter gene instead of a cytosolic protein. As was previously discussed, the use of a reporter protein that is secreted into the media does not allow one to determine the infectivity of these particles. Hence, it is likely that the yield of production of these particles is considerably lower than for the infectious virus, as we determined here, but it cannot be determined by measuring the bioluminescence of Gaussia luciferase. Furthermore, because not all the transfected cells contain both the replicon and helper plasmids, there is a fraction of particles that does not contain the replicon RNA. Nevertheless, the low yield of single-round infectious particles could be solved by either using a producer cell line that is not susceptible to CHIKV (which, given the wide tropism of this virus, is complicated to find) or by generating a stable cell line that continuously expresses the structural proteins. In fact, we are developing a stably transfected cell line that expresses the CHIKV structural proteins. In this system, when single-round infectious particles enter a stably transfected cell that is producing the structural proteins, it will package the replicon RNA. Therefore, with this system, we will be able to mimic the infectious process, but only when the single-round infectious particles enter cells that are stably expressing the structural proteins. This system will allow us to enrich the media with infectious particles.

## 5. Conclusions

Our results demonstrate that the generation of CHIKV infectious particles or of a CHIKV-derived RNA from a plasmid where transcription is driven by a Pol-II promoter and terminator requires that at least the 3′UTR be identical, as in the gRNA. The produced viral particles resulted in the expected cytopathic effect, plaques, and cytopathic vesicles, indicating that this plasmid results in particles that can mimic the infection of the wild-type virus. Further, the co-transfection of a replicon-independent helper plasmid and a replicon plasmid results in the assembly of single-round infectious particles that express the reporter gene in the target cells. However, this system has a low yield compared to the infectious clone.

The experiments described here are of great virological, biomedical, and biotechnological relevance. The pACNR-CHIKV plasmids can be used to produce infectious particles to study the virology of Alphaviruses, to determine the neutralizing of vaccine candidates, or to find antivirals that inhibit the CHIKV infectious cycle. The pACNR-Rep plasmid could be used in high-throughput approaches to find small molecules that inhibit the RdRp. The pVax-Helper plasmid could be used to understand the assembly process of CHIKV. Finally, this replicon/helper system will allow us to perform studies to understand the viral infectious cycle in a manner where we can decouple genome replication and assembly.

## Figures and Tables

**Figure 1 viruses-15-00132-f001:**
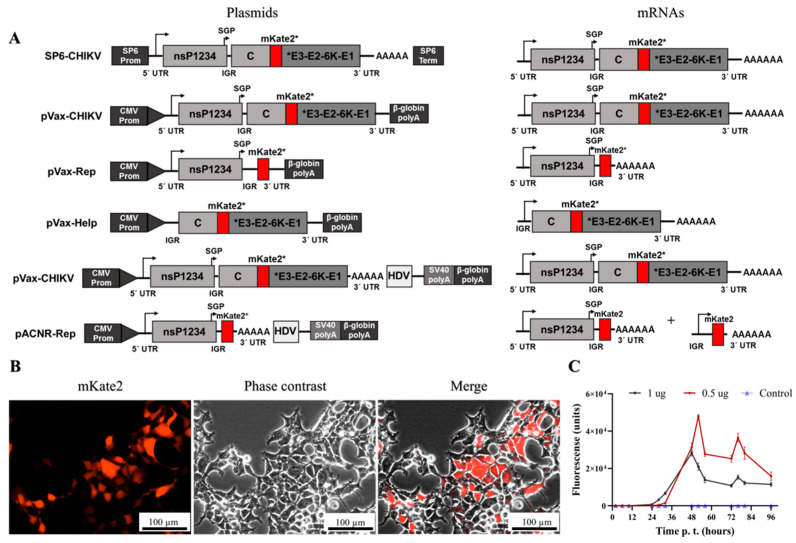
Construction and evaluation of an expression plasmid for CHIKV. (**A**) Schematic representation of the plasmids generated here. The SP6-CHIKV plasmid is used for in vitro transcription with SP6 polymerase and contains the full-length genome of the attenuated strain 181/25 of the Chikungunya virus. The genomic CHIKV was inserted into the pVax-CHIKV vector from which the human cytomegalovirus (CMV) immediate early enhancer and promoter sequences and βGH polyA derived from pVAX. The SV40 polyA and hepatitis delta virus (HDV) were inserted, and the genome was moved from the pVAX to the vector pACNR1811. The asterisk means that mKate2 is a reporter gene. (**B**) Micrographs of HEK-293T cells transfected with the pACNR-CHIKV plasmid in a fluorescent field (mKate2 channel), phase contrast, and a merge of both. (**C**) Kinetics of the expression of mKate2 from the plasmid pACNR-CHIKV. Bar, 100 µm.

**Figure 2 viruses-15-00132-f002:**
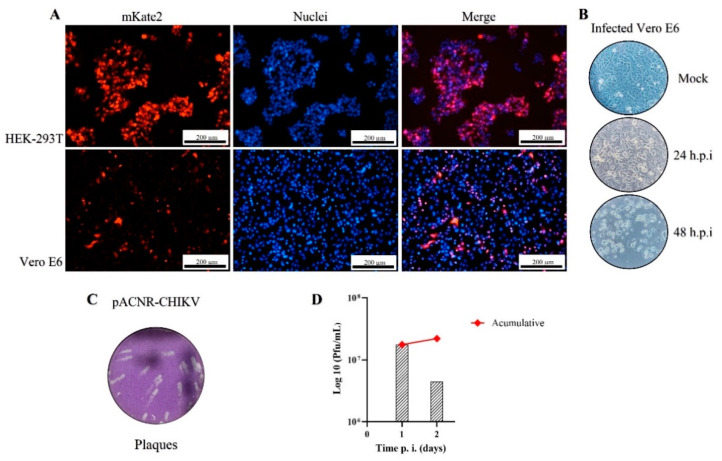
The Chikungunya virus plasmid pACNR-CHIKV produces a viral infection. (**A**) HEK-293T and Vero E6 cells infected with the supernatant of the producer (transfected) cells in fluorescent-field mKate2 expression (Texas Red filter), nuclei stain (DAPI filter), and merge at MOI of 1 for 24 h.p.i. (**B**) Cytopathic effects induced by the CHIKV infection in Vero E6 cells at MOI of 1. (**C**) CHIKV plaques on Vero E6 cells stained with crystal violet and (**D**) and virus titer measured as plaque-forming units (PFUs). Bar, 200 µm.

**Figure 3 viruses-15-00132-f003:**
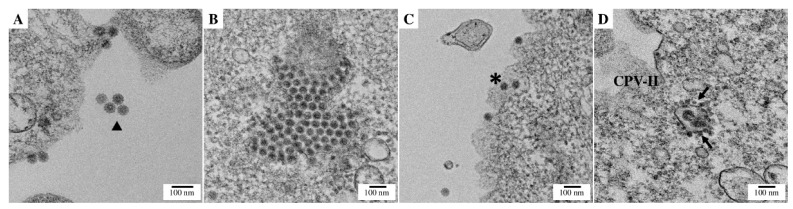
Chikungunya virus pACNR-CHIKV plasmid induces replication and assembly of viral infectious particles. Vero E6 cells infected with Chikungunya virus at a MOI of 1 were assessed at 24 h post-infection. TEM images show Chikungunya (**A**) four extracellular virions (▲), (**B**) a viral factory, (**C**) four virions budding from a membrane (*****), and (**D**) type-II cytopathic vacuoles (arrow, CPV-II). Bar, 100 nm.

**Figure 4 viruses-15-00132-f004:**
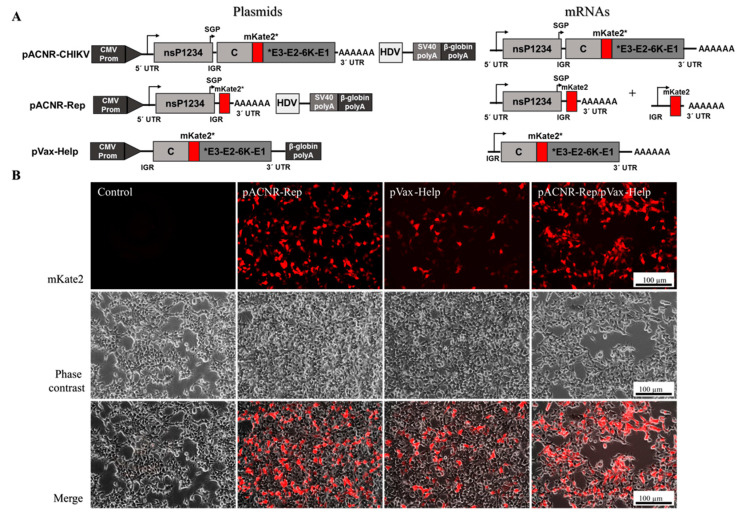
Generation and evaluation of replicon and helper vector system. (**A**) Constructions of pACNR-CHIKV, pACNR-Rep, and pVax-Help plasmids. The genomic CHIKV was inserted into the pACNR1811 vector from which the human cytomegalovirus (CMV) immediate early enhancer and promoter sequences, the SV40 polyA, and hepatitis delta virus (HDV) to generate pACNR-CHIKV. The structural proteins were removed to generate pACNR-Rep and derived from pVax-CHIKV and the nonstructural proteins were removed to generate pVax-Help. The asterisk means that mKate2 is a reporter gene. (**B**) HEK-293T cells transfected with pACNR-Rep, pACNR-Help, and co-transfected with both vectors after 48 h. Fluorescence micrographs with mKate2 expression (mKate2 channel), the morphology of cells in phase contrast, and a merge. Bar, 100 µm.

**Figure 5 viruses-15-00132-f005:**
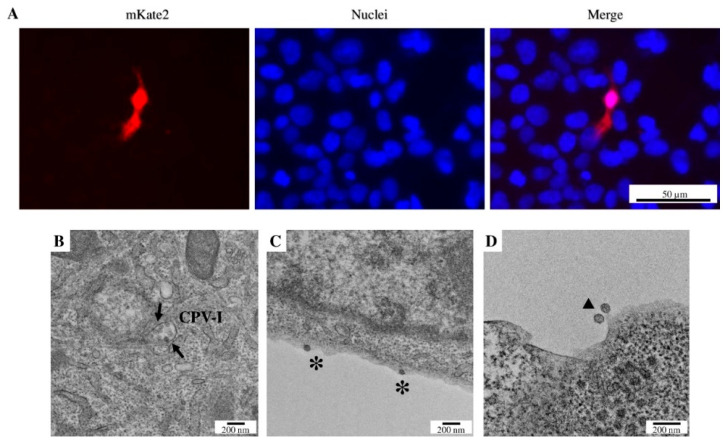
HEK-293T cells from the entry assay infected with the particles generated with the pACNR-Rep and pVax-Help system in (**A**) fluorescent-field mKate2 expression (Texas Red filter), nuclei stain (DAPI filter), and (**C**) overlap of mKate2 and nuclei micrographs, demonstrating that the particles generated by the co-transfection of both plasmids result in the expression of the gene of interest (mKate2) in the target cells. The bar indicates the same scale for all images, 50 μm. HEK-293T cells 48 h post-transfection with replicon and helper system were observed by transmission electron microscopy (TEM). (**B**) Cytopathic vacuole type 1 (arrow, CPV-I), (**C**) two virions budding from a membrane (*****), and (**D**) two extracellular virions (▲). Bar B–D, 200 nm.

## Data Availability

Annotated sequences of the plasmids generated here are available upon request.

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
