# Peer review of "Construction of a Chikungunya Virus, Replicon, and Helper Plasmids for Transfection of Mammalian Cells"

_viruses, 2022, doi:10.3390/v15010132_

Round 1

Reviewer 1 Report

I suggest rejecting the manuscript since to my opinion there is no novelty. Alphavirus-based plasmid DNA vectors are known since at least 24 years (see https://www.ncbi.nlm.nih.gov/pmc/articles/PMC6466081/ and references therein)

Authors are writing that they aimed to generate a „plasmid for expression in mammalian cells that contains the full-length genome of the attenuated strain 181/25 of the Chikungunya virus“ (line 94 & 95). DNA launched production of Chikungunya virus was described (https://www.ncbi.nlm.nih.gov/pmc/articles/PMC4038148/ ) as well as placing reporter genes into the viral genome (various paper). As a second goal they have „generated two more expression plasmids for mammalian cells that encode for either the ORF1 (replicon) or the ORF2 (Replicon-independent helper)“ (line 102-104), but also this is well known in the field. I have to admit that I am not aware of a paper describing the system based on the Chikungunya virus but to my opinion this is not something novel.

In conclusion, I would like to add that the purpose of the work is not clear to me.

Author Response

We appreciate the comments from the reviewer because they have helped us to improve the discussion. We partially agree with the reviewer that a Chikungunya clone carried by a plasmid for mammalian expression is not novel (please see Tretyakova et al (2014) The Journal of Infectious Diseases, 209, 12, 1882–1890, Suzukum et al (2021) Virology 552, 52-62, and Szurgot, et al (2020) Sci Rep 10, 21076). The rest of the published works with plasmids that carry the full length CHIKV genome are under a SP6 to T7 promoter for in vitro transcription, which limits their applicability (Kümmerer et al. (2012)  Journal of General Virology 93(9) 1991-1995, Scholte et al. (2013) PLOSOne PLOS ONE 8(8): e71047, Boussier et al (2022) PLOS ONE 15(11): e0241592, Tsetsarkin, et al. "andD. LVanlandingham (2006) Vector-Borne Zoonotic Dis 6).

            We believe that our article has the merits to be published because another key goal was achieved: a novel replicon/helper system was generated, comprising (i) a CMV-driven replicon with the 3´UTR identical to the WT virus, and (ii) a CMV-driven helper plasmid that does not require the presence of the replicon RNA to generate the structural proteins. This system is innovative since most Alphavirus replicon/helper system are based on helper RNAs that can only be amplified in the presence of the replicon RNA. However, we generated a replicon-independent helper plasmid. Furthermore, we have characterized the assembly of the single-round infectious particles by thin-section TEM, showing that, as in the WT virus, we can observe the assembly of the cytopathic vesicle type-1. This result shows that not only the replicon/helper system works, but its replication results in the same cellular modification as with the WT virus, which to the best of our knowledge, has not been shown for similar systems (please see the comment about the Utt et al. article). Nonetheless, to highlight the novelty of our article, we have cited in the discussion examples on the generation of either a replicon/helper system or just a replicon plasmid. As can be seen from the following explanation, none of these systems have the same characteristics as ours; hence, we believe that our work has a sufficient degree of novelty to be published.

To the best of our knowledge, there is only one article that describes the generation of a CHIKV replicon/helper system (Gläsker, et al. (2013) Virol J 10, 235 (2013). Their system has several key differences compared to ours: i) they used a plasmid for in vitro transcription using an SP6 polymerase, ii) the structural proteins are generated by using two different plasmids, one that codes for the capsid protein and a second one that codes for the glycoproteins, iii) The replicon RNA has  Gaussia luciferase as a reporter gene, and iv) the amplification of the helper RNAs depend on the presence of the RdRp. Therefore the replicon/helper system described in our manuscript is different since i) we do not need to in vitro transcribe the RNA as the sequences of interest are driven by a CMV promoter; ii) the generation of single-round infectious particles requires two plasmids rather than three, iii) the structural proteins are generated in the absence of the RdRp, iv) the reporter gene is a fluorescent protein that allows to directly identify the infected cells rather than measuring luciferase activity in the cell culture supernatant. This last point is important because, as we have shown, we can monitor in real-time the percentage of infected cells with no need for antibody labeling. It should be mentioned that the fact that we don’t need to use immunofluorescence to identify the infected cells is extremely important because, for the last three years, the Mexican Government has limited the importation of antibodies against viral proteins. We have tried to import antibodies against multiple proteins from CHIKV, DENV, and ZIKV, and even companies like Sigma-Aldrich/Merck and Thermofisher have been unable to do so. Therefore, the feature of using a fluorescent protein instead of an antibody to mark the infected cells is a major tool in our country.

Utt et al.( PLoS ONE 11(3): e0151616) generated a series of replicon plasmids that are driven either by a T7 polymerase promoter or CMV. Nonetheless, in order to determine the activity of the replicase they need to transfect the cells with at least two plasmids. The first plasmid encodes the RdRp and the second one contains the RdRp regulatory sequences flanking different reporter genes. Therefore our system has the advantage that if one wants to study the activity of the RdRp we only need to use one plasmid and not two. This results in an increased yield of cells that are expressing the reporter gene used to measure the activity of the RdRp. Also, their CMV-driven plasmid does not contain an HDV ribozyme at the 3´end, which could alter the efficiency of the replication process (as we have shown in this article). In fact, when they compared the expression of the reporter gene in the CMV- vs the T7-driven plasmids, it was clear that the CMV-driven plasmid has very little activity (if any) compared to the T7 plasmid. This is consistent with our finding; however, unlike in their work, we were able to solve the problem of the CMV plasmid by adding a ribozyme. Finally, the electron micrographs of the CPV-I in the Utt et al. article do not resemble the ones that have been reported for alphaviruses, while the ones we have observed are extremely similar, if not identical. Therefore, our system also generates replication vesicles identical to those obtained when infecting the cells with the full-length virus. Finally, Pohjala L et al. (PLoS ONE 6(12): e28923) also generated a similar system as the one from Utt et al., based on transfection with RNA transcripts rather than CMV-driven plasmids and thus is not comparable to what we have presented here.

We have addressed these issues in lines 131-175, 911-933, and 954-962

Reviewer 2 Report

Comments

Abstract

It needs to be clarified how the number of plasmids and if the reporter gene is in a different plasmid. This confusion is based on the way that it is written.

Furthermore, the biotechnological application is very repetitive, as is mentioned twice.

Introduction

It is essential to review the information about the storage of the carrier and vehicle as some do not need ultra-low temperature, and the seroprevalence information does not give the critical justification for establishing the CHIKV system. (Lines 68-74).

The authors do not show the importance of CHIV, describing the number of infections, distribution, etc. and how this system will help further studies in the CHIKV field.

Materials and Methods

I recommend summarising this section more. 

Results 

I suggest in this section describe the results as it looks like this part is a mix between methods and results.

Discussion

This section describes the results without looking at the justification and comparison data with the published literature. I recommend including it and discussing the results instead of describing them. 

Conclusion

The authors do not describe the importance of this system and the low recovery yield without giving possible solutions.

Questions: 

Why do the authors select the mKate2 reporter gene (Figure 1C)?

Why do the levels of mKate2 expression increase after 72hrs and decrease rapidly?

Did the authors use a different MOI 1 of 1 for the pACNR-CHIV results?

Can the authors add the plaques from the infection's kinetics, as in figure 2B?

Single-round VLPs? The vlps do not have genetic material; it just needs the structural genes for their production. Why did the authors use this term?

Can this system be used for animal challenge experiments?

Figure 2c. Could the authors discuss the variability of the diameter distribution?

VLP purification method. Does the CHIKV VLPs have a purification method after the harvesting by centrifugation?

Others

LINE 227-228. 

However, this method of transfection might not be optimal for biotechnological applications. 

This information is not part of the scope of this research. 

LINE 241-244 

The fact that transfection with pVax-Help in cells results in the expression mKate2 while there is no expression with pVax-CHIKV and pVax-Rep suggests a problem in the way viral RdRp polymerase recognizes the transcripts synthesized by a cellular polymerase.

I recommend describing this in the Discussion section and rewriting it.

MOI is mixed between abbreviated and full names. Can the authors change it to one style?

Author Response

We appreciate all comments, and we have done our best to solve each problem.

Abstract

1.- It needs to be clarified how the number of plasmids and if the reporter gene is in a different plasmid. This confusion is based on the way that it is written.

Furthermore, the biotechnological application is very repetitive, as is mentioned twice.

We have solved these issues.

Introduction

1.- It is essential to review the information about the storage of the carrier and vehicle as some do not need ultra-low temperature, and the seroprevalence information does not give the critical justification for establishing the CHIKV system. (Lines 68-74).

We have added the information about seroprevalence in lines 32-38 and 108-111.

We have modified 103-105 to remove to cost of the carriers, as the estimate we had was for the whole formulation and not only for the carrier.

2.- the authors do not show the importance of CHIV, describing the number of infections, distribution, etc., and how this system will help further studies in the CHIKV field.

We have added the epidemiological information in lines 32-38

We have addressed the second issue in lines 191-195

Materials and Methods

I recommend summarising this section more. 

We decided not to summarize this part as the other reviewer found it fine. The reason for not following this recommendation is to give enough information in case anyone wants to replicate our experiment.

Results 

1.- I suggest in this section describe the results as it looks like this part is a mix between methods and results.

We have followed this suggestion, and we have deleted the portions that look like methods.

Discussion

This section describes the results without looking at the justification and comparison data with the published literature. I recommend including it and discussing the results instead of describing them. 

We have modified this section according to this suggestion as well as per the suggestion of the other two reviewers.

Conclusion

The authors do not describe the importance of this system and the low recovery yield without giving possible solutions.

We have improved the conclusions accordingly to the suggestions.

Questions: 

1.- Why do the authors select the mKate2 reporter gene (Figure 1C)? -

The original clone that was donated to use contained this reporter gene which is very attractive because its fluorescence properties allow for its combination with other fluorescent probes (i.e., its excitation and emission range are on the far red of the fluorescence spectrum).

2.- Why do the levels of mKate2 expression increase after 72hrs and decrease rapidly?

This is due to virus-induced cell death. We have added this explanation on lines 389-391 and 570-573

3.- Did the authors use a different MOI 1 of 1 for the pACNR-CHIV results?

Yes, we only did infections at an MOI of 1.

4.- Can the authors add the plaques from the infection's kinetics, as in figure 2B?

The infection kinetics was not followed by plaque assay but by measuring the number of mKate2-positive cells.

5.-Single-round VLPs? The vlps do not have genetic material; it just needs the structural genes for their production. Why did the authors use this term?

Also, as per the recommendation of another reviewer, we have changed the term to single-round infectious particles.

6.- Can this system be used for animal challenge experiments?

Yes, it can be used.

7.- Figure 2c. Could the authors discuss the variability of the diameter distribution?

This distribution was done using the data from thin-section TEM. In this technique, we cannot be sure that all viral particles were sectioned through the middle part. However, this variation is consistent with what has been previously observed. We have added this observation to lines 587-590

8.- VLP purification method. Does the CHIKV VLPs have a purification method after the harvesting by centrifugation?

There are some methods published, but for this article, we have not purified the single-round infectious particles s because we are first focusing on optimizing the production, measured by the ability of the supernatant to infect naïve cells. However, the optimization of the production of the particles and their purification is part of a different project.

Others

1.- LINE 227-228. 

However, this method of transfection might not be optimal for biotechnological applications. 

This information is not part of the scope of this research. 

We agreed, and we have deleted this sentence.

2.- LINE 241-244 

The fact that transfection with pVax-Help in cells results in the expression mKate2 while there is no expression with pVax-CHIKV and pVax-Rep suggests a problem in the way viral RdRp polymerase recognizes the transcripts synthesized by a cellular polymerase.

I recommend describing this in the Discussion section and rewriting it.

We have followed this suggestion.

3.- MOI is mixed between abbreviated and full names. Can the authors change it to one style?

We have fixed this problem.

Reviewer 3 Report

In this manuscript, Colunga-Saucedo and colleagues report constructing a new chikungunya virus plasmid system, either for the full virus or for a replicon/helper system that can be transfected in a mammalian cell. The manuscript is reasonably well-written, but it could profit from more thorough proofreading, as multiple grammatical errors persist (see some of my comments below, probably not comprehensive). My main concern is about the current status of the research reported, which seems to be a 'work in progress' still, and its current state, it looks as if publication is too premature (see major comment below). 

Major comment

- The authors report here an incomplete work. While the preliminary results seem sound, and I do not doubt the success of their plasmid constructions, the system is not efficient yet. Indeed, the efficiency of infection produced by the double-transfected cells, arguably the paper's most interesting set of constructs for its potential application, is 'extremely low' (lines 430-431, line 458). This seriously limits the usability of these constructs for the goals the authors ultimately wish to achieve (lines 445-450). It is like having a car built that can only be driven at 5 km/h. It runs, but it is not helpful. The authors should first optimize the system, as they acknowledge themselves in the conclusion section (lines 458) and already provide possible ways to do so (lines 431-444). the authors even recognize they are working on it (lines 438-439). I would thus wait for this final step to be done and publish everything together. There is no hurry in publishing this paper first about the constructs.

Minor comment

- Lines 235-255: authors report that cells transfected with a high amount of DNA had at 48 h.p.t. fewer cells than those transfected with a lower amount. Do the authors have any hypothesis why this would be the case?

Typos

- Line 26: 'delivery...a self-replicating' => 'of' is missing

- Sentence of line 40-42 is confusing and has remaining typo, please rephrase

- Line 44: 'gene organization...Alphavirus" => 'of' or 'in' is missing

- Line 46: 'deletion...the' => 'of' is missing

- Line 51-52: 'deletion...the ORF2' => 'in' is missing

- Line 82: 'it production' => 'its production'

- Line 94: 'generation...a plasmid' => 'of' is missing

- Line 96: 'β-globing' => 'β-globin'

- Line 154: CMV appears here the first time, please provide complete name

- Line 169: 'all cells line' => 'all cell lines'

- Line 205: 'h.p.i' appears for the first time, please provide complete name

- Line 352: 'insolate' => 'isolate'?

- Line 368: 'after an the' => 'an' to be deleted

- Line 373: 'Indian Ocean, ,' => two commas, maybe something missing in between

- Line 397: 'demonstrated...the mechanism of' => 'that' is missing

- Line 400: 'at, 24 h.p.i. CHIKV' => 'at 24 h.p.i. with CHIKV, we were...'

- Line 409: 'structure 3’UTR that' => 'structure of the 3’UTR that'

Author Response

We appreciate the comments from the reviewer. However, we do not agree that this is a work in progress as it has defined goals; the generation of a series of CHIKV plasmids and their detailed characterization.  We have completed the goals of the articles, and thus we think is a finished product.

It should be mentioned that to the best of our knowledge, there is only one article about a CHIKV replicon/helper system (Gläsker, et al. Virol J 10, 235 (2013). In this article, they describe a three-plasmid system that allows for the generation of single-round infectious particles. However, they followed the infection in terms of Gluc luminescence in the culture´s supernatant; thus, they were not able to determine the infection efficiency. In contrast, in our approach, infected cells are directly detected by fluorescence. Nonetheless, thanks to our approach, we have realized that a two-plasmid system results in a low yield of infectious particles (when using a replicon/helper system).  This low yield was determined because we are looking at the number of infected cells rather than at the fluorescence of the media. Furthermore, when we talk about a low yield, we are actually referring to the yield of the single-round infectious particles compared to the wild-type virus (see lines 921-933).

The optimization of the replicon/helper system is outside the scope of this article. In fact, this optimization is a major project by itself we are working on. In order to optimize this system, we are generating a stably-transfected cell line that can be transfected with the replicon DNA. Also, we are trying to induce “superinfection inhibition,” a known effect for alphaviruses, to further increase the yield or production. Unfortunately, at moment we cannot discuss the details of this particular project as it will be patented.

Please see lines 493, 921-933, and 952-960

Minor points

We have modified the text accordingly.

1.- Line 32: bold type error ?

We have fixed the error.

2.- Line 86: original name pcDNA3.1

Done

3.- Line 86: pVAX1, please explain this vector at first appearance in your text, see line 128, 230

We have explained this plasmid in lines 91-93

4.- Line 326: one-round, is this single-round, look for consistency –

Done

5.- Fig 1AB, Fig 4AB, characters are too small and difficult to read in printouts

Done

6.- Line 300: is the term virus-like particle, correct? To my knowledge VLPs do not contain viral RNA or DNA. Is it a rec. virion or pseudotype ? –

VLPs can contain a genome, but it is a gray line, so we have changed it. We have changed this term in the article for single-round infectious particles.

7.- Line 209: TEM is not a good title of the section –

We have changed it by Ultra-structural analysis by thin-section Transmission Electron Microscopy

8.- Line 158: …with by PCR with the…. - we deleted the first with

Fixed

Reviewer 4 Report

The authors have developed a well-functioning system for gene transfer into cells using the chikununya virus. The publication is very well written and understandable. The experiments and statements are clear and very well supported by the data.

Minor points

Line 32: bold type error ?

Line 86: original name pcDNA3.1

Line 86: pVAX1, please explain this vector at first appearance in your text, see line 128, 230

Line 326: one-round, is this single-round, look for consistency

Fig 1AB, Fig 4AB, characters are too small and difficult to read in printouts

Line 300: is the term virus-like particle, correct? To my knowledge VLPs do not contain viral RNA or DNA. Is it a rec. virion or pseudotype ?

Line 209: TEM is not a good title of the section

Line 158: …with by PCR with the….

Author Response

1.- Lines 235-255: authors report that cells transfected with a high amount of DNA had at 48 h.p.t. fewer cells than those transfected with a lower amount. Do the authors have any hypothesis why this would be the case?

We do not have an exact answer to this question; we observed that a lower amount of DNA gives a higher transfection yield. We have observed this phenomenon for almost any transfection with Lipofectamine 3000®, it is possible that higher amounts of this reagent could be cytotoxic. Please see lines 576-578.

Typos

1.- Line 26: 'delivery...a self-replicating' => 'of' is missing

Done

2.- Sentence of line 40-42 is confusing and has remaining typo, please rephrase

Done

3.- Line 44: 'gene organization...Alphavirus" => 'of' or 'in' is missing

Done

4.- Line 46: 'deletion...the' => 'of' is missing

Done

5.- Line 51-52: 'deletion...the ORF2' => 'in' is missing

Done

6.- Line 82: 'it production' => 'its production'

Done

7.- Line 94: 'generation...a plasmid' => 'of' is missing – done

Done

7.- Line 96: 'β-globing' => 'β-globin'

Done

9.- Line 154: CMV appears here the first time, please provide complete name

The first time it appears is in line 127. We have added the correction there.

10.- Line 169: 'all cells line' => 'all cell lines'

Done

11.- Line 205: 'h.p.i' appears for the first time, please provide complete name

It was originally defined in the previous sentence

12.- Line 352: 'insolate' => 'isolate'?

Done

13.- Line 368: 'after an the' => 'an' to be deleted

Done

14.- Line 373: 'Indian Ocean, ,' => two commas, maybe something missing in between - - Done

15.- Line 397: 'demonstrated...the mechanism of' => 'that' is missing –

Done

16.- Line 400: 'at, 24 h.p.i. CHIKV' => 'at 24 h.p.i. with CHIKV, we were...' –

Done

17.- Line 409: 'structure 3’UTR that' => 'structure of the 3’UTR that' –

Done

Round 2

Reviewer 1 Report

Major

To my opinion, the work is of very limited originality. The revised version is more transparent in terms of what is already described in the literature. I strongly suggest to provide the annotated sequences of plasmids generated in this study (or at least sequences covered by schemas in Fig 1A) so other scientists, who want to use this technology, can get the relevant information in the manuscript.

Minors

Line 236: “Monolayers of cells were either infected with CHIKV at a of 1 or co-transfected with…” word MOI is missing between “…CHIKV at a” and “of 1…”

Line 447: Writing “Gläsker at al.” instead “Gläsker”

Line 448: “secreted into the supernatant” instead “secreted o the supernatant“

Line 428: Writing “Gläsker at al.” instead “Gläsker”

Author Response

We thank the reviewer for all the comments. We have submitted two a word and a pdf. The word file shows all the changes we have done using track changes and the PDF file shows the “simple markup” version. We have done this to make the reviewing process easier. Nonetheless, all the responses to the reviewers are based on the numbering from the word file. Also, we would like to point out that we deleted the last three sentences of the paragraph before the last one to reduce the a little bit the introduction as the suggestion from the reviewers considerably increase the length of this section.

1.- To my opinion, the work is of very limited originality. The revised version is more transparent in terms of what is already described in the literature.

  • We have improved the introduction to highlight the fact that, to the best of our knowledge, this is the only published CHIKV replicon/helper system based on CMV-driven plasmids that has a cytoplasmic fluorescent protein and that requires two plasmids to produce single-round infectious particles. The CHIKV replicon/system described in the introduction requires three plasmids, the reporter gene is secreted to the supernatant, and it requires in vitro transcribed RNA. Furthermore, the replicon system by Utt and coworkers requires two plasmids, rather than one, and their CMV-driven plasmids have almost no activity. The advantages of our system that give that makes this article original are the following: I) because our replicon/helper system requires only two plasmids rather than three (Gläsker) it increases the probability of a cell been transfected with the required molecules to produce single-round infectious particles, II) the use of a cytoplasmic reporter gene rather than one that is secreted to the supernatant allows to determine the percentage and identity of the infected cells; this is not possible with the already published system from Gläsker and co-workers, III) the study of the activity of the CHIKV RdRp by means of using a replicon RNA requires in our case one plasmid, while the one from Utt and co-workers require two; this difference increases the yield of transfected cells and thus is a more attractive system, and IV) unlike the CMV-driven replicon plasmid ours has a very high biological activity because we have corrected the 3´UTR, thus we do not need to use in vitro transcribed RNA to transfect cells. We believe that these differences with the current CHIKV replicon/helper and replicon plasmids merit the publication of this article. These issues have been addressed both in the introduction and discussion.

2.- I strongly suggest to provide the annotated sequences of plasmids generated in this study (or at least sequences covered by schemas in Fig 1A) so other scientists, who want to use this technology, can get the relevant information in the manuscript.

            - We have included this information in the supplementary information and stated that the electronic files with the sequences are freely available upon request, see lines 498-500 (section.  2.1)

Minors

1.- Line 236: “Monolayers of cells were either infected with CHIKV at a of 1 or co-transfected with…” word MOI is missing between “…CHIKV at a” and “of 1…”

-           Fixed

2.- Line 447: Writing “Gläsker at al.” instead “Gläsker”

-           Fixed

3.- Line 448: “secreted into the supernatant” instead “secreted o the supernatant“

  • Fixed

4.- Line 428: Writing “Gläsker at al.” instead “Gläsker”

  • Fixed

Reviewer 2 Report

General comments from each section

Introduction

The authors try to explain the importance of using the Alphavirus replicon/helper system for different therapies, but the introduction does not give enough facts to consider it compared with the other described systems. I recommend improving this if the introduction has this as an objective. 

Materials and Methods

As my first report, I strongly recommend summarising this section more. Even if the other research groups want to reproduce it, they will follow their protocols just using the critical information that is difficult to get from this publication's material and methods. 

Example: the full commercial name of the competent bacteria or plasmids analysed by Sanger sequencing, as this other information is not necessary. 

Results

The section still with information relevant to the material and methods. The figures are not well explained, and the findings from each result are not shown quickly. Also, the discussion section has essential information about the results section. Therefore, I recommend to re-write this section.

For example:

Transfection of HEK-293T cells with pVax-CHIK (data not shown) and pVax-Rep (Figure S1) did not result in the detection of the reporter gene mKate2. However, transfection with pVax-Help did result in the expression of mKate2 (Figure S1).

The authors have to focus on the relevant information regarding the results that are part of the scope of this publication. 

Discussion

This section has relevant information for the results section, or other information is repetitive in both sections. The discussion seems more like explaining the results instead of its discussion.

For example, the attenuated strain CHIK 181/clone 25 derived from an isolate Thailand (This data is not described in the results section)

Questions or observations

Could you include the reference about the low probability of viral genome integration from the described vectors from LINE 65?

The low-temperature requirement for a lipid system is still not fully precise, as this system can be stored at room temperature. 

Do the authors have experimental controls of the PRNT assay?

Do the producer cells have the entry factor for the single-round particles? This question concerns the proposed scenario. 

Could the authors add information about the entry factors on the studied cell lines for the CHIKV system?

Author Response

We thank the reviewer for all the comments. We have submitted two a word and a pdf. The word file shows all the changes we have done using track changes and the PDF file shows the “simple markup” version. We have done this to make the reviewing process easier. Nonetheless, all the responses to the reviewers are based on the numbering from the word file. Also, we would like to point out that we deleted the last three sentences of the paragraph before the last one to reduce a little bit the introduction as the suggestion from the reviewers considerably increase the length of this section.

Introduction

The authors try to explain the importance of using the Alphavirus replicon/helper system for different therapies, but the introduction does not give enough facts to consider it compared with the other described systems. I recommend improving this if the introduction has this as an objective. 

- We have fixed these problems please see lines 108 to 128.

Materials and Methods

As my first report, I strongly recommend summarising this section more. Even if the other research groups want to reproduce it, they will follow their protocols just using the critical information that is difficult to get from this publication's material and methods. 

Example: the full commercial name of the competent bacteria or plasmids analysed by Sanger sequencing, as this other information is not necessary. 

- We have summarized as much as possible.  

Results

1.- The section still with information relevant to the material and methods.

- We have fixed these problems 

2.- The figures are not well explained, and the findings from each result are not shown quickly.

- We have fixed these problems 

3.- Also, the discussion section has essential information about the results section.

Therefore, I recommend to re-write this section.

 For example:

Transfection of HEK-293T cells with pVax-CHIK (data not shown) and pVax-Rep (Figure S1) did not result in the detection of the reporter gene mKate2. However, transfection with pVax-Help did result in the expression of mKate2 (Figure S1).

  • We have fixed these problems in the discussion section. However, the point that the reviewer is trying to make is confusing. This is the beginning of the results section and thus we are describing the results of those transfections.

4.-  The authors have to focus on the relevant information regarding the results that are part of the scope of this publication. 

- We have fixed these problems in the introduction (lines 108 to 128) and we have improved discussion.

Discussion

This section has relevant information for the results section, or other information is repetitive in both sections. The discussion seems more like explaining the results instead of its discussion. For example, the attenuated strain CHIK 181/clone 25 derived from an isolate Thailand (This data is not described in the results section)

 - We have fixed these problems

Questions or observations

1.- Could you include the reference about the low probability of viral genome integration from the described vectors from LINE 65?

  • We have added the reference about lentiviral vectors, but we have removed the sentence about episomal DNA as this if used correctly can be an advantage.

2.- The low-temperature requirement for a lipid system is still not fully precise, as this system can be stored at room temperature. 

  • Here we refer to Pfizer and Moderna’s handling protocol for the COVID-19 vaccines where the samples have to be kept at -80ºC for transport before reconstitution. We have fixed this in the introduction.

3.- Do the authors have experimental controls of the PRNT assay?

  • We have not carried out any PRNT assays as we do not have antibodies against CHIKV. If the reviewer refers to the plaque assays, yes indeed we always had negative controls that yield in no plaques. We have added this in lines 572 and 572

4.- Do the producer cells have the entry factor for the single-round particles? This question concerns the proposed scenario. 

  • Yes, they have it. This is shown in figure 1C where cells were transfected and 24 hours later the cells were washed to remove the supernatant. Also, this is shown in figure 2A where cells were infected with viral particles, and we see the expression of the reporter gene.
  • Furthermore, Suzki et al have previously shown that HEK-293T cells are susceptible and permissible for CHIKV infection. This has been added to the discussion in lines 996-999
  1. Could the authors add information about the entry factors on the studied cell lines for the CHIKV system?
  • We have address this in lines 115-117 and 996-999

Reviewer 3 Report

My initial assessment of the manuscript has not been addressed by the authors in a satisfactory fashion. The work is still too preliminary, and I do not believe the optimization is out of scope when a paper presents a new 'tool' (here the plasmid system). This work is still a work in progress. If it is meant to be patented, then why publish?

Author Response

We thank the reviewer for his comments. We have submitted two a word and a pdf. The word file shows all the changes we have done using track changes and the PDF file shows the “simple markup” version. We have done this to make the reviewing process easier. Nonetheless, all the responses to the reviewers are based on the numbering from the word file. Also, we would like to point out that we deleted the last three sentences of the paragraph before the last one to reduce a little bit the introduction as the suggestion from the reviewers considerably increase the length of this section.

1.- My initial assessment of the manuscript has not been addressed by the authors in a satisfactory fashion. The work is still too preliminary, and I do not believe the optimization is out of scope when a paper presents a new 'tool' (here the plasmid system). This work is still a work in progress.

  • We disagree with this comment. As has been explained before, and as the other reviewers have noted, the aim of this article was to generate a series of CMV-driven plasmids for the production of fully infectious and single-round infectious CHIKV particles. Furthermore, we generated a unique replicon/helper system that requires only two plasmids, and that allows us to determine the percentage and identity of infected cells. Unlike what has been published this system does not require generating in vitro transcribed RNA. We would like to point out that this goal has not been achieved by others and thus it represents a significant improvement to the current CHIKV systems. Our results justify this as a complete and round project. We do acknowledge the fact that the system can be improved. However, the generation of a stable cell line that increases the yield of single-round infectious particles is a major project that we are currently working on that should stand on its own and that is outside the goals of this manuscript

2.-  If it is meant to be patented, then why publish?

  • We want to patent the production system that improves the yield of single-round infectious particles because this technology has the potential to be transferred to an industrial partner. However, the fact that we want to patent this system does not preclude us from publishing it after the patent is submitted, which should occur by the end of 2023.